# Bridging the Research Gap—A Framework for Assessing Entrepreneurial Competencies Based on Self-Esteem and Self-Efficacy

**Jaana Seikkula-Leino** [1,2,*] and **Maria Salomaa** [1,3]

1   RDI & Business Operations, Tampere University of Applied Sciences, 33520 Tampere, Finland; maria.salomaa@tuni.fi
2   Department of Education, Faculty of Human Sciences, Mid-Sweden University, SE-851 70 Sundsvall, Sweden
3   Lincoln International Business School, University of Lincoln, Lincoln LN6 7TS, UK
*   Correspondence: jaana.seikkula-leino@tuni.fi

**Abstract:** This research contributes to the growing discussion on entrepreneurial competencies from a multidisciplinary point of view, integrating elements of entrepreneurship research, education sciences, and psychology. Despite many efforts to develop and utilize different entrepreneurial constructs in entrepreneurship education, there is still a lack of theoretical framework for systematic development and measurement validation related to entrepreneurial competencies. This paper aims to widen the theoretical and conceptual discussion on entrepreneurial competencies by stressing the roles of self-esteem and self-efficacy. The study aims to contribute to the academic discussion (1) by addressing this research gap through a literature-based analysis on how entrepreneurial competencies, self-efficacy and self-esteem relate to each other; and (2) by presenting a conceptual framework (ENTself) for further development of entrepreneurship education. The results of the study reflect how self-esteem and self-efficacy are connected, and how they can be aligned with entrepreneurial competencies. We argue that a systematic, theory-based approach to further research on entrepreneurial competencies, based on the proposed framework, is needed for a broader understanding and facilitation of entrepreneurship education. Also, the development of assessment tools adapted from ENTself is suggested for conducting future research of the framework and its validation.

**Keywords:** assessment; entrepreneurial competencies; entrepreneurship education; self-efficacy; self-esteem

## 1. Introduction

Fostering entrepreneurship and entrepreneurial behavior can be key drivers of societal, environmental, and economic change [1]. They can, as an example, facilitate the transition towards environmentally sustainable societies whilst contributing to innovative activities. Furthermore, they enhance job-creation. However, a systematic development of dynamic entrepreneurial competencies requires significant investments on individuals' skills and capabilities to release their full potential [2]. Thus, new solutions for promoting innovation, creativity, and problem solving, aligned with environmental, social, and economic well-being, have been continuously explored [3,4]. In parallel, implementation of entrepreneurship education activities has been recommended in order to develop these needed competencies [5], which has led to significant public investments in entrepreneurship education programs [6]. This suggests that one of the future challenges is to encourage entrepreneurship education providers to clearly delineate the theoretical foundations of the provided courses and programs, and to both track and adequately measure the impact of the educational programs. Therefore, entrepreneurial educators should be encouraged to adopt new educational innovations and processes, which have empirically proven to lead to positive outcomes [7,8].



Bosman, Grard and Roegiers [9] argue that in the field of entrepreneurship education, an individual, competency-based approach has become the most common method to design and structure training programs and courses. It digresses from what entrepreneurs are towards what they do, hence emphasizing the role of competencies needed for entrepreneurial behaviour. As Chandler and Jansen [10] conclude, the entrepreneurial competencies are fundamental in performing and succeeding well in different areas of life. Subsequently, the academic discussion have been reinforced by new elements, such as integrating the concept of 'competence' to the learning processes [11–13]. This approach argues that competencies should not be viewed as inputs, outputs or processes, but rather as context-dependent processes of learning. However, adopting such a holistic perspective requires a thorough definition of all the related key concepts for solid theoretical underpinning [13], and an explicit description of what is meant by 'entrepreneurial behaviour' and 'entrepreneurial learning'. This is also crucial for responding to the actual needs of entrepreneurship for teachers and educators, as well as for developing entrepreneurship education as a whole. This is supported by Devici's and Seikkula-Leino's [14] research, in which they examine and review 76 studies conducted on entrepreneurship education in the context of teacher education. They conclude that the efficient implementation of entrepreneurship education in practice requires further promotion and an in-depth understanding of what is entrepreneurial learning and what are the students' entrepreneurial characteristics, such as entrepreneurialism. At the same time—and to a greater extent—there is a need to assess entrepreneurship education and its impact [15]. The development of meaningful metrics is also the subject of much interest [16].

The European Entrepreneurship Competence Framework (EntreComp) is one of the EU's responses to support common understanding and widespread integration of entrepreneurship, within and across education systems, promoting the development of entrepreneurial competencies and entrepreneurialism in societies [17,18]. However, there is no doubt that the exploration of entrepreneurship education and entrepreneurial competencies requires new research openings. Therefore, in our paper, we discuss how interdisciplinary approaches have not been sufficiently taken into account in entrepreneurship education and entrepreneurial competencies for further investigating these issues.

This article will build on previous studies by seeking to develop a profound understanding of entrepreneurial learning and behaviour in order to promote students' entrepreneurial competencies. The aim is to formulate an assessment framework for entrepreneurial competencies based on previous studies by Borba [19,20], Aho [21–23], Seikkula-Leino [24] and Seikkula-Leino & Salomaa [25], and Nevalainen et al. [17]. We argue that Borba's theory could be useful in generating an in-depth understanding of entrepreneurialism. Moreover, Borba's work will be further developed by integrating different aspects from psychology, entrepreneurship and entrepreneurship education research for creating new approaches and future practical interventions to the development of entrepreneurial competencies. Our results will also provide new perspectives to the theoretical discussion of entrepreneurship education, being that previous studies on entrepreneurship education and entrepreneurial behaviour are mainly focused on empirical survey findings. Although empirical studies are relevant, their findings often point out the need to create a solid theoretical understanding of entrepreneurial behaviour behind practical interventions related to entrepreneurship education [6,26–28].

To sum up, despite many efforts to develop and utilize different entrepreneurial constructs in entrepreneurship education, the state-of-the-art literature on human psychological behavior and entrepreneurial competencies currently lacks a theoretical framework for systematic development and measurement validation related to entrepreneurial competencies.

Thus, this paper aims to widen the theoretical and conceptual discussion on entrepreneurial competencies. Our focus is on the concepts of self-esteem and self-efficacy and how they are connected to entrepreneurial competencies. By focusing on these psychological aspects of human behaviour, we seek to investigate how these concepts relate

to entrepreneurial behaviour. Firstly, entrepreneurial behaviour and learning, self-esteem and self-efficacy, will be presented and their connections to each other will be summarized. Secondly, we focus on Borba's [19] self-esteem theory. Thirdly, we will summarize the current state of understanding related to entrepreneurial competencies based on a systematic literature review. Fourthly, in the discussion part, we combine the key aspects of the previous sections by proposing a new framework, ENTself—Assessment Framework of Entrepreneurial Competencies Integrating self-esteem and self-efficacy. Finally, the potential limitations of this theoretical framework and suggestions for future research and practices will be discussed.

## 2. Literature Review

In this section, we will discuss Bandura's theory of self-efficacy [29–32] and Borba's theory of self-esteem [19,20]. The latter is considered to be not only a theory of self-esteem, but it has noticeable connections to the theories of self-efficacy. Therefore, we have chosen to focus on these two interrelated key concepts of psychology and educational research. In terms of entrepreneurship education, entrepreneurial learning and behaviour, there is a need to develop better understanding on the ground, components and situations in which entrepreneurial learning could be evoked in real-life settings (e.g., by developing practical activities, tasks, learning environments as well as reliable and validated assessment tools). Thus, Borba's self-esteem theory provides a fruitful starting point as it does not only focus on self-esteem and its formation as a traditional theoretical discussion', but it is also a very practice-oriented approach, which is rather rare in the field of psychology. Her work involves assessment questionnaires and tasks to use with students, and this has been attractive for teachers and other people working in real classroom settings—unlike many entrepreneurship education studies, which do not provide novel approaches to underlying theories or concrete strategies to be employed for educational interventions [7,33,34].

As integration of different aspects of self-efficacy has been seen as a crucial in boosting entrepreneurial intentions in the field of entrepreneurship research, we consider Borba's work to be highly relevant in the development of the conceptual framework of entrepreneurial learning. Moreover, as stated earlier, the research of entrepreneurship education is largely focused on empirical research, whereas theoretical and conceptual studies are still absent in the field. This article strives to address this gap by deriving a theoretical framework for entrepreneurial competencies by integrating Bandura's and Borba's previous work. Next, these two theoretical approaches will be presented in more detail, and their connections to entrepreneurial learning (e.g., risk taking, problem solving, learning from failures, opportunity creation and creativity) will be discussed in the context of entrepreneurship education and entrepreneurial learning.

### 2.1. Entrepreneurship Education Promoting Entrepreneurial Competencies

Entrepreneurship education research builds mainly on the conceptual understanding of entrepreneurship and learning, containing various aspects of psychology and learning constructs to which we will delve into more detail in this study. As Gibb [35] states, entrepreneurship education is about learning for entrepreneurship, learning about entrepreneurship and learning through entrepreneurship. It has been suggested that entrepreneurship education should be thus considered as the method, practice and content of learning [36–39]. According to Pepin [40], entrepreneurship education's key educational dimensions are: (1) entrepreneurship education is an object of studying, involving teaching entrepreneurship more as a discipline and aiming towards, for example, starting up businesses or venture creation; (2) entrepreneurship education is about entrepreneurial pedagogy in which, e.g., entrepreneurship is as a tool for acquiring skills to deal with real-world challenges and act and cope in the complex society and working life. Thus, its objective is more towards developing students' entrepreneurial mindset.

In this article, we refer simultaneously to both mainstream concepts of entrepreneurial education and entrepreneurship education while investigating the factors that manifest

through entrepreneurial competencies and behaviour. The learning outcomes from entrepreneurship education have been claimed to include several aspects. Entrepreneurship education introduces entrepreneurship as a career choice, it supports the entrepreneurial way of seeing and doing things, and it characterizes a particular way of teaching and learning [41,42]. Furthermore, it encourages students to strive to take more responsibility for themselves, their actions and learning [43,44]. In other words, entrepreneurship education can support students' feeling of their internal locus of control. As a learning outcome, students can try to more persistently achieve their goals, to be creative, to discover existing opportunities, and to cope with increasingly complicated society in general. Thus, entrepreneurial education involves the development of attitudes, behaviors, skills and attributes applied individually and/or collectively to help individuals and organizations of all kinds to create, cope with, and embrace change and innovation [43,45]. Entrepreneurship education also promotes competencies which are needed for entering into working life [14,37,38]. This is also stated in the European EntreComp framework, which is a policy-driven initiative to establish a common language for entrepreneurial competencies to bridge education and work, and to understand entrepreneurship as a key competence [18].

From the entrepreneurship research point of view, entrepreneurship education can also relate to opportunity creation according to mainstream innovation theorists such as Schumpeter [46] and Kirzner [47]. The Schumpeterian theory claims that opportunities will emerge through new combinations of existing resources, whereas Kirzner emphasizes the 'holes' in the markets which, in terms of resources, could be used more effectively. These two dominating mainstream approaches to entrepreneurship research create a theoretical basis for understanding the entrepreneurial processes. They are often presented as competing approaches, although according to Nielsen et al. [48], these two theories can be considered as complementary as they mutually support our understanding of entrepreneurial thinking and behaviour.

Entrepreneurship research has also borrowed from psychological aspects of entrepreneurial and enterprising processes [49]. Researchers of entrepreneurship have proposed several intention models, such as combining personal and contextual factors as well as self-efficacy [50–53]. As an example, a rather widely used framework to analyze the impact of entrepreneurship education (EE) is Ajzen's 'Theory of Planned Behaviour', which focuses on individual's entrepreneurial intentions [54] and EE's positive effect on the desirability and feasibility of starting a business [55]. However, as stated earlier, entrepreneurship education can also have other goals than starting up a company.

There is a consensus that entrepreneurial processes are not often linear [56] but are instead iterative processes. This means that attitudes, intentions and behavior are dynamically interrelated [57] and may vary drastically [58]. The same dynamics run through experimental learning, in which learning is perceived as a process, in which new knowledge is generated through the transformation of experience [59]. In this study the concept of experience, which is used in educational research, fosters our knowledge development in entrepreneurship education [60]. Furthermore, according to the socio-constructivist approach, new knowledge is indeed created and revised in a particular social context. These theoretical points of view from education research also grounds learning in entrepreneurial and entrepreneurship education.

However, there is not a full consensus on the theoretical underpinnings, nor the impact or outcomes of entrepreneurial learning. This suggests that there is a need for widening the discussion on entrepreneurship education to promoting entrepreneurial competencies. The discussion should also be expanded to be more interdisciplinary, as students can no longer be educated in subject orientated academic silos [1]. Thus, we need to develop a more in-depth understanding of entrepreneurial learning in different research and practical settings, as well as to identify which factors have an impact on human entrepreneurial behavior. Therefore, we seek to investigate what can be learned from the previous studies framing competencies towards a new conceptualization.

## 2.2. Self-Esteem, Self-Efficacy and Other Key Concepts Related to Them

Research of entrepreneurship states that raising entrepreneurial efficacies will raise perceptions of venture feasibility, thus increasing the perception of opportunity [54,61] and entrepreneurial intentions [62]. Additionally, according to Wilson, Kickul and Marlino [63] self-efficacy may play an important role in shaping (or limiting) perceived career options as early as in the middle and high school years. Moreover, Arora et al. [64], Laguna [65], Staniewski & Awruk [66]; Piperopoulos & Dimov [67] and Neto et al. [26] found in their studies that self-efficacy predicts entrepreneurial behavior.

Self-esteem is closely linked to self-concept. Self-concept is "your idea(s)" and beliefs about yourself, including individual attributes of who and what this personal self is [68–71]. The main difference between self-concept and self-esteem is that the latter is also connected to the individual emotional factors. For example, any piece of information about one's self may be incorporated into the self-concept. Therefore, this information affects self-esteem once it takes on a value judgement: How would I consider myself? Is it good or bad about me? High self-esteem denotes thinking well of oneself; and, in theory, low self-esteem is the opposite of high self-esteem. However, this issue is not straightforward. According to many studies, hardly anyone is convinced that they are entirely "bad" people. Thus the low self-esteem is the absence of more positive beliefs than the presence of negative thoughts about the self. Also, to keep in mind, high self-esteem and narcissism are not quite the same thing. Most narcissists have high self-esteem. However, many people with high self-esteem are not narcissists. This is also reflected in behavior. For example, people with high self-esteem do not worry about what others think. They trust themselves and their actions, and can also deal with failures. Unlike narcissists, it is essential that they feel superior to others without failures and mistakes and can mistreat other people to achieve their own social goals to "keep their face" [68].

The effects of self-esteem on achievement have attracted considerable attention from the scientific and nonscientific public. There is a widespread belief that a positive "self-view" leads to higher achievement. However, we cannot take this for granted. For example, high self-esteem does not necessarily lead to high school achievements, for example, and vice versa [24,72–74]. However, there also many studies supporting the positive correlation between high self-esteem and positive achievements, e.g., [24,72–75]. This kind of positive connection between self-esteem and outcomes is also supported by Rabeh & Neila's [76] research, in which they investigated the effect of self-esteem, entrepreneurship education, and entrepreneurial tradition of the family on the entrepreneurial intention among students. However, it should be noted that behavior and performance development is significantly influenced by the social context [77] and, for example, the learning environment created [74]. Therefore, for example, achievements are not only based on internal factors.

As opposed to self-esteem, self-concept is a more objective description of oneself. Self-concept includes, for instance, social, physical and emotional self-concepts as well as a learning self-concept [69–71]. Individual self-esteem indicates whether an individual is aware of him/herself, whether he/she knows him/herself and how he/she values him/herself. Self-esteem also emphasizes self-value and knowledge of it. Thus, a realistic concept of self-image does not differ much from good self-esteem. Self-value that has affective and evaluative aspects determines more than a mere self-concept; in self-image, the cognitive side is emphasized, whereas self-esteem is always associated, besides evaluation, with strong subjective experiences at an emotional level whilst self-image has a more objective approach [20,23,24].

### 2.2.1. Self-Efficacy by Bandura

Self-efficacy is a belief that he/she can accomplish a particular activity [29]. It also differs from locus of control by relating to competence in special situations. Thus, it is more than general cross-situational beliefs about control. Thinking and actions are organized through self-organization, proactivity, self-regulation, and self-reflection. People are not simply onlookers of their behavior, but they are contributors to their life circumstances—not

just products of them [31]. Bandura [29] explains that there are four sources of efficacy beliefs;

1.  Mastering a success: for example, in a task or controlling an environment, it will build self-belief in that area, whereas a failure will undermine that efficacy belief. To have a resilient sense of self-efficacy requires experiences of overcoming obstacles through effort and perseverance.
2.  Vicarious Experiences: seeing people similar to ourselves succeed by their sustained effort raises our beliefs that we too possess the capabilities to master the activities needed for success in some area. This second source of self-efficacy comes from our observation of people around us, especially people we consider as role models.
3.  Verbal Persuasion: influential people in our lives, such as parents, teachers and managers, can strengthen our beliefs that we have what it takes to succeed. Being persuaded that we possess the capabilities to master certain activities means that we are more likely to put in the effort and sustain it when problems are to come.
4.  Emotional & Physiological States: this source is about how you judge your self-efficacy. Negative emotions, such as depression or stress, can dampen one's confidence in his/her capabilities.

Self-efficacy also plays a major role in organization development. As Bandura [32] explains, students' beliefs in their efficacy regulate their learning, motivation and mastering accomplishments. Self-regulation refers to an individual's active participation in his or her own learning process. It is the process through which self-generated thoughts, emotions, and actions are planned and systematically adapted as necessary to one's learning and motivation [13].

Ongoing self-appraisals and self-regulation are the key dynamic determinants in these self-system processes of students' affective experiences and cognitive learning. Promoting self-awareness, positive self-appraisals and efficient self-regulation will empower students' learning and problem solving, for example, to consciously act on debilitating affective responses. They may then choose to 'fine tune' the role of their affective responses in learning and problem-solving processes [78]. As Op't Eyende et al. [79] stress, teaching students how to solve problems, (e.g., which is usual in entrepreneurship education), then implies that we also have to teach them how to cope effectively with feelings of frustration or sometimes anger. In other words, allowing space for negative emotions might be an educational goal from a cognitive, as well as motivational, point of view. This approach is also supported by a study by Kyrö et al. [13], in which they investigated meta processes of entrepreneurial and enterprising learning, and the dialogue between cognitive, conative and affective constructs. According to their results, stressing affective factors and their self-regulation could have more potential in education and entrepreneurial learning than we are aware of.

Moreover, teachers' beliefs of their personal efficacy both motivate and promote learning. They have an impact on the types of learning environments they create for their students, as well as on the level of academic progress they accomplish with their students [80]. Furthermore, faculty and schools' beliefs of their collective instructional efficacy contribute significantly to their level of academic achievements and other outcomes. Thus, self-efficacy is not only an individual process, but a phenomenon formulated both through individuals and groups which can stimulate entrepreneurial competencies on both individual and collective levels.

Bandura [30] has also explained differences between self-efficacy and self-esteem. According to him, self-esteem is a judgement of self-worth, whereas self-efficacy is a judgement of capabilities. This divide suggests that there are major differences between these two concepts. Since self-esteem is presented in this paper from a different viewpoint in comparison to Bandura, we will be return to this in the following sections and discuss the integrated links between self-efficacy and self-esteem.

Bandura et al. [81] have also included self-efficacy as one of a variety of sociocognitive influences on the career aspirations of children, and found that academic self-efficacy had



the strongest direct effect. While the relationship between self-efficacy and career choice has been well established in the career theory literature, most studies have not included specific career options around entrepreneurship [63]. However, there are also studies that support the correlation of self-efficacy and career intentions, specifically in the realm of entrepreneurship e.g., [63,82].

### 2.2.2. Self-Esteem by Borba

Borba bases her research on Burns [69,70] and Shavelson et al.'s [71] studies, as well as Reasoner's theory [19,83,84]. She divides self-esteem into five different components, which can help individuals to shape the internal image of self as a subjective experience, which may not be consistent with other people's views of himself/herself [19,22].

According to Borba [19] students with high self-esteem usually perform well in the following areas: Security, Selfhood, Collaboration, Mission and Competence.

**Security:** feeling of security or emotional security: 'I feel safe and trust my friends and teachers. I dare to try out new solutions. I know what is expected from me'. An individual with a stable basic social security is able to assess his/her abilities. He/she is also able to function in different and changing environments and is able to receive varying information from these situations. For example, if a student has problems with basic social security, they may isolate themselves and avoid social situations. In addition, the student confirms things multiple times and may need external confirmation, such as a teacher's validation, for things they do. On the other hand, problems with basic social security may lead to a situation in which rules and instructions are disregarded. Therefore, in distressing situations, defenses are also relied on. [21,24]

**Selfhood:** everyone is special, individuality or harmonious identity: '*Who am I*'; A freedom to shape an individual identity: an individual is able to appreciate themselves. Such an individual knows who he is and what he believes in. He perceives himself as eligible, and he has a realistic understanding of his roles and qualities. Such an individual is able to receive constructive criticism because he recognizes his qualities and abilities. Being aware of oneself also creates the feeling of security and the individual has the capacity to praise and encourage others, see, e.g., [19,20]. Problems with selfness cause, among other things, negative self-description and oversensitivity to criticism. A person is not able to recognize his role nor his abilities. Such a student wants to be inconspicuous or correspondingly draws special attention. The individual does not recognize his abilities and is not able utilize them. He may also underrate the significance of assessment [22,23].

**Affiliation:** collaboration skills, the feeling of belonging or the feeling of unity: '*I feel that I am being accepted as a part of the community*'. The third dimension concerns interaction and attitude towards others. It is typical for social cohesion that an individual identifies with a community and a group. Consequently, understanding one's uniqueness is not only relevant for an individual's development. It is important, for example, that a student feels accepted in studies-related group situations. A strong feeling of togetherness arises when a student feels accepted and appreciated by people who the student appreciates. Their social contacts are positive. They dare to venture into new friendships and interactive situations. An individual with a strong feeling of togetherness is cooperative and masters sympathy [22–24].

**Mission:** goal orientated behaviour: '*I know what I believe in and where I go. I can set goals for my actions*'. Students, who are orientated towards goals, perceive their life as meaningful. They know the objectives and direction. These students also set goals for themselves and accomplish them. Problems and hardships do not discourage the student. The individual is able to find alternative and creative solutions and minimize problems in problematic situations [19,20]. If a student has poor mission and goal orientated behaviour, they are not able to get motivated in performance situations, but rather they get bored. They are not able to see alternatives in their actions. Feelings of weakness and helplessness are typical. Such a student does not set goals, gets discouraged easily, avoids responsibilities and is not

able to make decisions. Often, tasks remain unfinished. In addition, indetermination often describes life [22–24].

**Competence:** 'I feel that I am an efficient student, who can also influence the actions of others. I solve problems. They serve my studies. I look for new opportunities and also take risks. I achieve my goals'. The feeling of competence arises from the confidence that the student is able to reach the set goals, overcome problems that he faces, and by pursuing his dreams. This is an individual who perceives themselves as competent, is able to seek alternative solution models, solve problems, create and use their capacity effectively. The individual is aware of their strengths, but is also able to accept short-comings. The feeling of competence also prevents total failure, because mistakes are perceived as a part of learning. A student who perceives themselves as competent feels from time to time that they are successful. Such a student is glad to share their ideas and opinions [19,20,22–24,84]. If an individual's feeling of competence is weak, they are unwilling to share their ideas and thoughts with others. They do not want to take risks nor try again. This also slows down the development of creativity. Such an individual is helpless and perceives themselves as useless even in such areas in which they are skillful. If the individual performs well, they underrate their performance or believe that their success originates from coincidence [21,22].

At first, an individual's self-esteem forms from three elements; these are basic security, selfhood and affiliation. The environment has an important role in their development. When these three elements develop, the individual can form a more specific and realistic picture of themselves. As a result, mission and competency are formed. Thus, the importance of external control gradually decreases, and the individual does not need to rely up on others' opinions, but rather the individual becomes internally-driven. Such internal drive is called self-empowerment. An individual who strengthens his/herself is able to detect alternative solutions to problems. This improves, for example, problem-solving capabilities, creativity, innovation, taking risks and enables setting more specific and challenging goals, through which life can be perceived as more meaningful. When an individual is concentrated on recognizing his/her strengths, accomplishing goals and is aware of different alternatives, flaws and weaknesses can be accepted without them significantly weakening his/her self-esteem, but mistakes and hardships become tools of learning. Also, when an individual strengthens his/herself and accomplishes the set goals, they do not need to seek other people's approval. They know themselves, and that they have succeeded, which helps them to try new things and take risks [19,24,85].

Borba [19,20] also stresses the self-esteem of an organization or/and a group: In a community with members with high self-esteem, they all fell accepted, and it is easier to communicate in a group. It feels good to make plans about common goals and results are also achieved. A strong group can also cope with setbacks, and it is willing to create solutions to solve problems. In an organization or in a group, we can even gain new energy from setbacks and create new opportunities. Feelings about our own work and about our self-esteem in a community are even a key to our well-being in school.

### 2.2.3. A Summary of Borba's and Bandura's Views: Similarities and Differences

Comparing and summarizing Borba's and Bandura's views of positive self-belief development, there are both similarities and differences. Both researchers emphasize goal-orientated behaviour, competence and the meaning of community, although Borba focuses more on interaction development and Bandura on role models. Bandura also refers to the emotional and psychological state of a person, whilst Borba stresses the components of selfhood and security. Both approaches have been tested to be relevant in education. In previous studies, Borba's [19] theory has been tested several ways, indicating this research approach may be valid in terms of indicating motivation, causal attributions and academic results, e.g., [21–24,38,85]. For example, in Seikkula-Leino's research [8], in which she studied under- and overachievers in bilingual education, it was found out that academic self-image [69–71,85] is correlated with the feeling of competence [19], school achieve-

ments of mathematics and languages, and how students' motivation, in terms of 'sense of control—causal attributions' [85] can have causal relationships with self-esteem and school results. Later, Borba's approach was utilized in the research of entrepreneurship education, e.g., in terms of studying entrepreneurial team behavior [86], entrepreneurial staff competencies [25], and designing a measurement tool for entrepreneurship education [6,16].

Currently, there are very few published studies presenting the role of self-esteem in the context of entrepreneurship research, even when it is a frequently studied variable in many other psychological domains, such as self-efficacy studies [65,87]. However, preliminary connections have been examined in Staniewski & Awruk's [66] study, which highlighted how entrepreneurial success is related to general self-esteem, self-efficacy and various other psychological factors. Unlike in psychology and science (e.g., medicine), self-efficacy and self-esteem have arguably been the most researched constructs of human self-evaluations that are closely related also to their mental well-being [88–90].

Therefore, we may estimate that entrepreneurial behavior could be a construct that involves both self-esteem and self-efficacy. However, first there is a need to develop its theoretical construct for further testing and measurement validation.

However, we have begun this work. For example, in our research related to the development of an entrepreneurial organization [25] and pedagogy development for entrepreneurial team learning [17], we have made preliminary use of these theoretical starting points, and we have created preliminary indicators. According to results, the framework would appear to be relevant for studying and developing entrepreneurial learning. Admittedly, in this concept paper, we delve into the rationale of the framework and deepen it from the discursive point of view.

In the next section, the chosen methodology for the framework development—an interdisciplinary literature review—will be presented, after which we will examine if a combination of Borba's and Bandura's approaches allows developing a novel construct to assess entrepreneurial competencies by integrating education, psychology and entrepreneurship research.

## 3. Method: Literature Search

A systematic literature review was undertaken in February 2021 in line with the research aims outlined in the introduction, and the following research question that was posed:

How are both self-efficacy and self-esteem studied in the context of entrepreneurial behaviour and/or learning, and/or entrepreneurship, and/or entrepreneurship education?

The literature search was conducted in Scopus and the Web of Science database using the search strings 'self-efficacy', 'self-esteem', 'entrepreneurial behaviour', 'entrepreneurship', 'entrepreneurship education' and 'entrepreneurial competencies'. This search rendered 916 results. During prereading, titles and abstracts were analyzed and articles related to all concepts were selected, leading to a total number of 41 articles. Only research papers or other work published by a peer-review process were included. The following attributes of articles were analyzed to produce an initial categorization of articles and to decide whether an article was to be considered for further analysis: aim of the article/research focus, language, theoretical contribution, research methods, and type of results (qualitative, quantitative). Research was only accepted for further analysis if they provided a concrete theoretical contribution to this study. Based on these criteria, 24 peer-reviewed studies were included in the analysis. Articles were coded deductively and inductively using a concept-centric approach [91] to analyze the key findings on entrepreneurial behaviour/entrepreneurship, entrepreneurial competencies, self-esteem and self-efficacy, and creating a theoretical basis supporting development of entrepreneurial educational practices. The information provided by the latest research papers was integrated into 'traditional research' in the field starting from early 1900s.

## 4. Results

As previously explained, former studies on entrepreneurial learning and behaviour have not explicitly covered the theoretical underpinnings behind the development of entrepreneurial competencies. Therefore, we aimed to develop a summarized theoretical basis for an in-depth understanding of entrepreneurial competencies. Moreover, we emphasize opening self-esteem and self-efficacy research in this area. As much of our previous research has been based on the development of measurement tools for entrepreneurial behavior or entrepreneurship education, e.g., [6,16], it is also stressed how the model derived from literature could support the development of practices and assessment tools for entrepreneurial education.

The Table 1 illustrates how entrepreneurship research emphasizes self-efficacy, educational psychology from a self-esteem research point of view, entrepreneurship education research, theoretical openings for learning (as experimental, student-centered and social learning), and the principles guiding the development of a practice-supported learning theory model, which could be summed up towards the conceptualization of a new construct in the field. The table shows how entrepreneurship (according to entrepreneurship research) is based on self-efficacy and even self-esteem, thus promoting entrepreneurial intentions.

On the other hand, opportunity-oriented (even future-orientated and proactive) thinking is also a key feature of entrepreneurialism. Furthermore, strong foundations in educational psychology emerge to emphasize self-esteem and efficacy in this context. In entrepreneurship education research, entrepreneurship is seen as an issue that develops in for, about, and through entrepreneurship in which, according to the research of learning, experimental and *'hands on' learning* play an important role. Recently, the discussions in entrepreneurship education have also stressed the development of competencies (e.g., entrepreneurial competencies) by which an individual may cope in society, e.g., in working life. On the other hand, it is also possible to focus more intensely on working life, such as by starting a company and acting as an entrepreneur. In general, when we set out to take forward the development of a theoretical framework in education, it would be wise to consider how theory could also serve practice, such as the development of assessment in the field, e.g., [33,34,37,38,92–94].

**Table 1.** A literature based framework demonstrating connections from different disciplines and approaches to understand entrepreneurial competencies and their assessment in education.

| Defining Connections from Different Desciplines and Approaches to Understand and Develop Entrepreneurial Competencies in Education: Literature in Education/Literature in Entrepreneurship Education and Entrepreneurship | Literature in Entrepreneurship Education | Literature in Entrepreneurship | What Are the Connections from Different Disciplines and Educational Approaches? |
|---|---|---|---|
| Literature from learning theories Learning by doing: hands-on approach for learning, meaning learners must interact with their environment in order to adapt and learn. (e.g., [95,96]) Learner-centered and constructivist-based learning (e.g., [97,98]) Learning is a social process in which a learner may reach his or her full potential (e.g., [77]) The learner is responsible for developing his/her learning and learning environment (e.g., [99]) | Learning for, about and through entrepreneurship (e.g., [35]) Entrepreneurship education as an object and as a pedagogical approach (e.g., [40]) Developing competencies for entering into working life and creating new business; Realizing entrepreneurship education in practice by assessment (e.g., [14,25,37,38]) | Self-efficacy research supporting the idea of boosting entrepreneurial intentions and its connection to self-esteem (e.g., [26,31,32,50–53]) Opportunity creation (e.g., [46–48]) | Education reformers have, throughout decades and even centuries always emphasized learner-centredness and the learner's self-activity. Genuine learning also takes place in a social context where the learner also has responsibilities and freedoms. These contexts are also outside of a formal school that enhances the integration into working life and society. Furthermore, the learner takes action to develop new ideas, thus promoting, e.g., entrepreneurship in society. |

**Table 1.** *Cont.*

| Defining Connections from Different Desciplines and Approaches to Understand and Develop Entrepreneurial Competencies in Education: Literature in Education/Literature in Entrepreneurship Education and Entrepreneurship | Literature in Entrepreneurship Education | Literature in Entrepreneurship | What Are the Connections from Different Disciplines and Educational Approaches? |
|---|---|---|---|
| **Literature from other educational psychology** Self-esteem research supporting education practices and their assessment & integrating the aspects of self-efficacy research (e.g., [19,20,24,85]) | | | Entrepreneurship research focuses on self-efficacy. On the other hand, the same issue has been addressed in educational psychology through self-esteem research. Different disciplines emphasize the individual's thinking, self-belief, thus also influencing one's actions, e.g., in creating new opportunities. |
| **Literature supporting a theory creation enhancing assessment in education** (e.g., [33,34,37,38,92–94]) | | | In general, research in education, entrepreneurship education, and entrepreneurship often focus on developing and testing new practices. To create appropriate, e.g., teaching practices, one must first understand what they should include. After this, indicators based on theory or other frameworks can be developed. Thus we see (a) how, for example, teaching practices work and (b) how valid the theory or a framework is and (c) consider possible needs for further development. |

Therefore, we present ENTself—a Fremework for Assessing Entrepreneurial Competencies, in the discussion of this paper based on this described summary of how we see the foundation of entrepreneurial competencies.

## 5. Discussion

Figure 1 illustrates how empowering entrepreneurial learning is based on the previously summarized literature, thus supporting its future utilization in the assessment of entrepreneurial competencies.

In the ENTself framework, we adapted Borba's five core components of self-esteem: security, selfhood, affiliation, mission and competence. Moreover, we integrated the aspects of entrepreneurship and working life into this model as future (career) paths. In entrepreneurship education, this can also refer to employment readiness, orientation towards further studies, and opportunities and working life in general, although generating new business ideas is an inherent part of entrepreneurship. The latter can also mean generating new ideas for the development of the workplace of students. This can also include opportunity creation [46,47]. Furthermore, Borba's self-esteem approach has been selected from the point of view of stressing self-efficacy that has been considered to be crucial for boosting entrepreneurial intentions in the field of entrepreneurship research [26,31,32,50–53]. Moreover, Borba's approach has previously been theoretically and methodically triangulated several ways, e.g., [24].

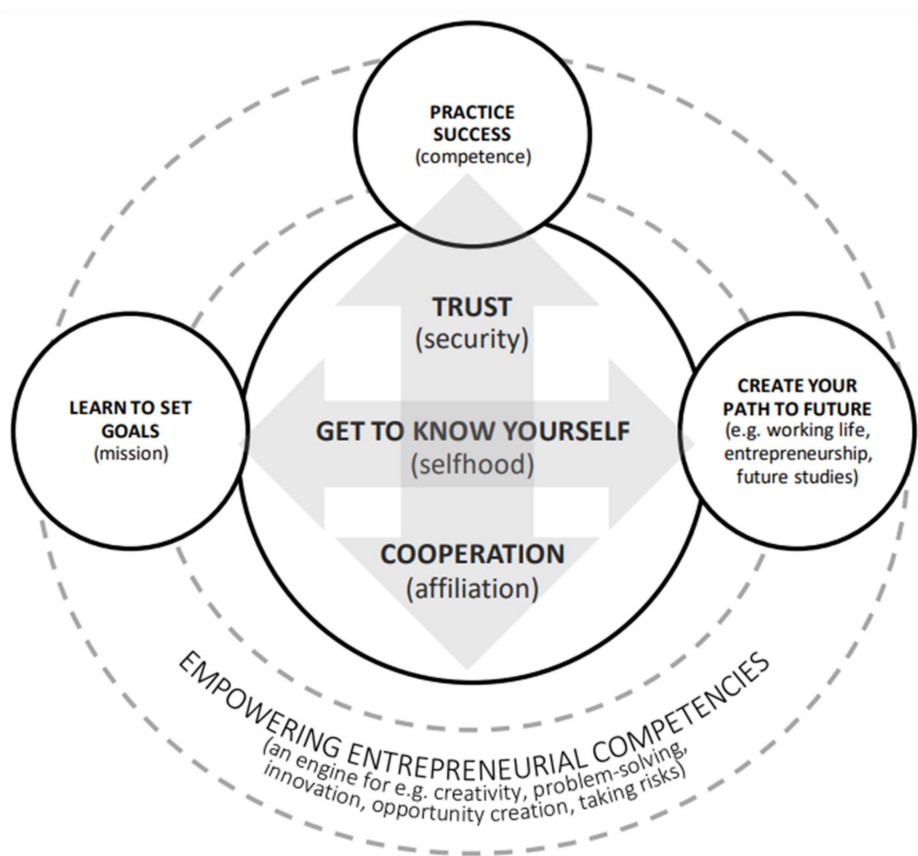

**Figure 1.** ENTself—a Framework for Assessing Entrepreneurial Competencies.

Next, we revised the six core components to be more explicit for both students and teachers, while still stressing Borba's work. The ground of entrepreneurial learning involves the following components: (1) Trust; (2) Get to know yourself; (3) Cooperation; (4) Learn to set goals; (5) Practice success; and (6) Create your path to future (e.g., working life, entrepreneurship, future studies). The model is based on Borba's idea of self-empowerment, which is also formed in the activities of the group, (e.g., Bandura [29]). Social processes for learners are also essential to empower the existing potential of both individuals and groups [77,95,96]. Entrepreneurial learning proceeds experiential learning [59], student-centered education [97,98], learning by doing [95,96], and allowing and encouraging students to take responsibility and have freedom to decide for themselves. They may design their learning environments and actions to enhance their learning [99]. At first, an individual's self-esteem forms from three elements; these are basic security, selfhood and affiliation. The environment has an important role in their development. When these first three elements develop (see Figure 1: "Trust, Get to know yourself, Cooperation") the individual forms a more specific and realistic picture of himself. As a result, goal setting and success will improve. Thus the importance of external control gradually decreases, and the individual does not need to rely upon other's opinions. The individual becomes internally driven. As explained in the literature review, such internal drive is called self-empowerment, which supports the development of entrepreneurial competencies involving creativity, problem solving, risk taking, and other kinds of entrepreneurial behavior. However, it is mentioned that these processes are complex and linear estimations, and conclusions may not be taken as self-evident.

How do these six approaches could be seen in entrepreneurship education? As an example, in terms of cooperation, a student may find new creative ways to develop friendships. Or in terms of developing self-trust, a student will create solutions for a problem which may arise when presenting new ideas aloud. Furthermore, all these created

approaches reflect Gibb's idea [35] of learning about, for and through entrepreneurship. Entrepreneurial behavior is more or less about a way of thinking, acting, and a way of working every day. It is not just about establishing a company. Moreover, this model is not only for students' learning. Future research address questions the link between the teacher and student learning [100]. It is also about teacher or other trainer education and relationships between research practices [33,34,37,38,92–94]. Therefore, it is suggested that, in the future, it also combines different research approaches from different actors, point of view in a very practical way, in order to promote practices in classroom settings.

However, it should be noted that even though the key concepts and their relationship have been defined, this research is still decidedly tentative and thus needs a great deal of effort to reach the level of theory. In order to achieve more learning interventions, both formal and informal, based on the model, perhaps our interventions may have culture-bound specifics that might affect the findings and change the dynamics of the model. Furthermore, a tested conceptual approach may still be missing some critical aspects. Through continuing research, we can further develop our model by studying other relevant literature as well as upcoming data. In order to create a reliable and valid construct of entrepreneurial competencies, further assessment will be needed for its validation.

This work has already begun in practice. For example, in the SKILLOON (www.skilloon.com (accessed on 16 August 2021)) learning environment, which is an official education concept of Education Finland and supported by the Finnish National Board of Education, we have taken steps to develop AI-based metrics based on the framework. In addition, by utilizing Borba's [19,20] hands-on research-based materials for education and other good teaching practices related to the framework and entrepreneurial learning, we have formed the basis for using this framework in education and its research. In the future, our research will also focus on arguing how these pragmatic concepts have been utilized in the SKILLOON learning environment.

## 6. Conclusions

This paper aimed to widen the theoretical and conceptual discussion on entrepreneurial competencies. We have focused on the concepts of self-esteem and self-efficacy and how they are connected to the development of entrepreneurial competencies. We combine the key aspects of both concepts by proposing a new framework, ENTself—Assessment Framework of Entrepreneurial Competencies Integrating self-esteem and self-efficacy. In our study, we have shown how self-esteem and efficacy are interrelated with entrepreneurial competencies. However, there are many different definitions of for these concepts. For example, there are many different nuances in the definition of self-esteem, which poses some challenges to deriving unequivocal results based on the research literature.

Thus, the value of this research is in deepening and clarifying our understanding of entrepreneurial competencies by presenting the ENTself framework, which is based on interdisciplinary research drawing from psychology, education and entrepreneurship, and by discussing its potential utilization in entrepreneurship education. Entrepreneurship education providers need to clearly delineate the theoretical foundations of their courses and programs, and to both track and adequately measure their impact [7,33,34]. Although the framework may yet have many shortcomings when operationalized into practice, it is nevertheless a novel starting point for future development of indicators and research-based educational testing of the model. Also, we are already taking steps toward creating suitable assessment tools and practices, and our future research will focus on their effectiveness. Moreover, it is becoming more and more essential to consider whether we are doing 'the right things' in entrepreneurship education—or do we only think so? Is our framework still lacking elements that can only be detected in the implementation phase? Yet only by creating and researching different frameworks and conceptual approaches for entrepreneurship education can we be able to deepen our understanding on what entrepreneurship entails, how it could be promoted, and how education can support entrepreneurship through various practices.

The current challenges of entrepreneurship education are also largely due to the fact that it has been 'invented' for the business studies. However, today entrepreneurship education includes developing competencies that enable individuals and communities to cope with serious problems and to develop societies in a sustainable way. Therefore, entrepreneurship education is much more than creating a startup. Thus, entrepreneurship education and research on entrepreneurial competencies should be strengthened in a more comprehensive and interdisciplinary way in the future. In fact, education reformists have emphasized entrepreneurial learning throughout the ages, without using the concept of entrepreneurship education. Therefore, we could also use the ideas of practitioners to reform pedagogy in order to develop entrepreneurship education more widely. This would also open up its value more strongly for different disciplines.

Although our study is only the first pilot study in its field, we see much added value in this opening, despite its limitations. This study contributes to the development of a growing pool of knowledge in research of entrepreneurial competencies to finally achieve a wider generalizability of the key findings, for example in promoting entrepreneurship education in societies. By providing new insights to the understanding of entrepreneurial competencies and creating different frameworks, for example, for its evaluation, we are able to put entrepreneurship education much better into practice; as an example we have presented a preliminary theoretical framework on which our emerging metrics and practical solutions are based on, namely the current version of the SKILLOON environment.

In the future, entrepreneurship education could have more to offer in the development of educational practices in general. As mentioned before, its scope and potential impact are not always understood sufficiently. Entrepreneurship education is psychologically and socially based on the fact that a person is a whole being. We have approached the creation of entrepreneurial competencies from this point of view. To be entrepreneurial, he or she must have an understanding of himself or herself and of other people as well. In addition, he/she must believe in him/herself and his/her possibilities. Cooperation is an inherent part of it all. Actions need to be targeted, at least to some extent, to make a change. Instead of underachieving, an individual with profound entrepreneurial competencies aims for success, although risks and failures are part of this process in which creativity also plays an important role. Future thinking, such as an orientation to working life and entrepreneurship, is also a part of how individuals relate in a societal context. Thus entrepreneurial competencies are relevant for all aspects of everyday life, such as formal education, informal education, daily situations, working life, and social connections and thoughts. Entrepreneurial individuals, having entrepreneurial competencies, create an entrepreneurial society in cooperation, and vice versa; entrepreneurial communities may empower entrepreneurial individuals that have courage and strength to aim for their dreams, realize innovative actions together, and build better societies.

**Author Contributions:** Conceptualization, J.S.-L. and M.S.; methodology, J.S.-L.; validation, J.S.-L.; formal analysis, J.S.-L.; writing—original draft preparation, J.S.-L. and M.S.; writing—review and editing, J.S.-L. and M.S. All authors have read and agreed to the published version of the manuscript.

**Funding:** This research received no external funding.

**Institutional Review Board Statement:** Not applicable.

**Informed Consent Statement:** Not applicable.

**Data Availability Statement:** There is no data in this study.

**Conflicts of Interest:** The authors declare no conflict of interest.

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
