# Peer review of "Bridging the Research Gap—A Framework for Assessing Entrepreneurial Competencies Based on Self-Esteem and Self-Efficacy"

_education, doi:10.3390/educsci11100572_

Round 1
Reviewer 1 Report
This is a very interesting and relevant research. The article is written with professional confidence and shows good knowledge in the area of entrepreneurship education. However I have two main concerns and some minor issues listed below where the main concernes are also presented.
A. I think creativity is not given enough attention and role if this is truly to be a "practice-orientated assessment framework for entrepreneurial competencies".
B. I also do not think it is clear HOW this framework could be used in practice.
Both issues can be addressed without revamping the article and should be a moderate challenge for the authors.
Issues to address or consider:
- In line 12 – starts with Ee – and seems half finished. At least Ee does not make sense.
- The pink highlights on pages 6 & 7 indicate that creativity is an important factor that needs to be more prominent in the model to truely exemplify WHAT should be developed in entrepreneurship education. Creativity is such an important psychological part of a person´s experiences and influences their self-image that it seems strange to leave it out of a model about EE.
- Line 486-487 you claim we will able to put entrepreneurship education much better into practice – i still am not convinced, how will the model be applied in practice? HOW will it help put EE into practice?
- And then you add: better understand what should be developed in entrepreneurship education. And this WHAT is missing the very important part, important competence in EE that is creativity and its emergences (e.g. problem solving).
- Line 502 developed by self-reflection – they are not only developed by self reflection, certainly also developed by applying creativity in practice.
- I can not see Getting to know your self or Practice success without execising Creativity – I think it is important to include that element in the model – whether it is presented as a core, a thread or an overarching element. If I were to scrutinize my EE courses with a model I would certainly not like to miss assessing whether creativity was applied.
- In paragraph 460-467 you do indicate the role of creativity – however it is not visible in the model, not even indistinctively.
- Language is overall good but, in some instances, there are minor faults – and one major fault in the research question as the verb must be second in order to be a question –
line 109: crusial – should be crucial.
Line 114: this article strives to …. This is strange wording
107 & 473, class room is written as two words, should be classroom - A lead researcher on collective teacher (school) efficacy is Tschannen-Moran, it would be strong to include a citation to her research supporting Bandura´s findings in lines 230-231: Tschannen-Moran, M. & Barr, M. (2004). Fostering student achievement: The relationship between collective teacher efficacy and student achievement. Leadership and Policy in Schools, 3, 187-207.
- The aim of the paper was to formulate a practice-orientated assessment framework for entrepreneurial competencies – it is not clear how this framework would be applied and it is missing an integral part. What does it have that makes it more appealing than the EntreComp framework? I think giving such a rationale would help to convince the reader of this model´s/framework´s value.
- Surely this claim does not apply for this article: The dataset generated for this study will not be made publicly available because of the sensitive nature of the questions. All study participants were assured that the data will remain confidential and will not be shared.

Author Response
This is a very interesting and relevant research. The article is written with professional confidence and shows good knowledge in the area of entrepreneurship education. However I have two main concerns and some minor issues listed below where the main concerns are also presented.
A. I think creativity is not given enough attention and role if this is truly to be a "practice-orientated assessment framework for entrepreneurial competencies".
Thank you. We highly appreciate that you highlight promoting creativity in this context. It needs to be more explicit. We have described below the way in which we have revised our paper, keeping this issue in our mind (p. 7, parag. 3, 5, 6; p. 8 parag. 1; p. 12 parag. 1; p. 11, Figure 1; p. 12 parag. 1; p. 13 parag. 4)
B. I also do not think it is clear HOW this framework could be used in practice.
Thank you. In the table 1 and in the discussion part (p.12, parag. 4) we have developed this part from the assessment point of view. However, in our future papers we will concentrate how this framework could be applied in more detail in education.
Both issues can be addressed without revamping the article and should be a moderate challenge for the authors.
Thank you. We have tried our best to address these issues in the current version.
Issues to address or consider:
- In line 12 – starts with Ee – and seems half finished. At least Ee does not make sense.
Thank you for pointing this out. We have revised the whole paper.
- The pink highlights on pages 6 & 7 indicate that creativity is an important factor that needs to be more prominent in the model to truely exemplify WHAT should be developed in entrepreneurship education. Creativity is such an important psychological part of a person´s experiences and influences their self-image that it seems strange to leave it out of a model about EE.
Yes, we have revised our paper in several parts to integrate more explicitly creativity into it. Furthermore, we have revised Figure 1.
3. Line 486-487 you claim we will able to put entrepreneurship education much better into practice – i still am not convinced, how will the model be applied in practice? HOW will it help put EE into practice?
We have tried to clarify how our framework can be beneficial for developing practices. Please see discussion and conclusion (p. 12, parag, 4; p. 13. parag. 1)
4. And then you add: better understand what should be developed in entrepreneurship education. And this WHAT is missing the very important part, important competence in EE that is creativity and its emergences (e.g. problem solving).
Please, see our comments above.
5. Line 502 developed by self-reflection – they are not only developed by self reflection, certainly also developed by applying creativity in practice.
Thank you for this comment, we have deleted this part.
6.I can not see Getting to know your self or Practice success without execising Creativity – I think it is important to include that element in the model – whether it is presented as a core, a thread or an overarching element. If I were to scrutinize my EE courses with a model I would certainly not like to miss assessing whether creativity was applied.
Thank you for this relevant point, we fully agree. Please see our comments above.
7.In paragraph 460-467 you do indicate the role of creativity – however it is not visible in the model, not even indistinctively.
Thank you, we have revised this model.
8. Language is overall good but, in some instances, there are minor faults – and one major fault in the research question as the verb must be second in order to be a question –
line 109: crusial – should be crucial.
Line 114: this article strives to …. This is strange wording
107 & 473, class room is written as two words, should be classroom
Thank you for these important remarks. We have proofread the revised paper.
9. A lead researcher on collective teacher (school) efficacy is Tschannen-Moran, it would be strong to include a citation to her research supporting Bandura´s findings in lines 230-231: Tschannen-Moran, M. & Barr, M. (2004). Fostering student achievement: The relationship between collective teacher efficacy and student achievement. Leadership and Policy in Schools, 3, 187-207.
Thank you. We have integrated this interesting reference to the revised version.
10. The aim of the paper was to formulate a practice-orientated assessment framework for entrepreneurial competencies – it is not clear how this framework would be applied and it is missing an integral part.
As described above, we have clarified this issue.
What does it have that makes it more appealing than the EntreComp framework? I think giving such a rationale would help to convince the reader of this model´s/framework´s value.
Thank you. We have clarified how our framework is connected with EntreComp. Moreover, we describe the benefits of using our framework (p. 2, parag. 3).
11. Surely this claim does not apply for this article: The dataset generated for this study will not be made publicly available because of the sensitive nature of the questions. All study participants were assured that the data will remain confidential and will not be shared.
That is indeed current, thank you for this remark.
Reviewer 2 Report
The authors make an important point regarding the importance of theoretical rationale for entrepreneurial competencies in addition to empirical evidence. Self-esteem and self-efficacy may be relevant contributors to such a rationale. Unfortunately, neither the arguments made nor literature reviewed are sufficient to constitute a theoretical rationale as intended. Overall, the authors have established a laudable goal, but they have not accomplished that goal in the present version of the manuscript. Specific suggestions can be found below.
- Numerous spelling errors make aspects of the manuscript difficult to understand, particularly towards the beginning (e.g., crusial, emphising).
- The comparison of different self-efficacy theories would be clearer if it were formatted more like the summary that appears after each theory is described, rather than as a list of concepts for each theory separately.
- Some ideas are oversimplified or possibly misunderstood, e.g., Kolb wrote about “experiential” learning, not “experimental learning” (Table 1). Similarly, phrases like “Unlike in the psychology and medicine research, self-efficacy and self-esteem have arguably been the mostly researched constructs of people’s self-evaluations that are closely related also to their mental health” (Lines 363-365) are unclear.
- Self-esteem is a controversial topic in psychology, because many studies have found that it is not linked to outcomes one might expect based on common understandings of the term. A more nuanced view of self-esteem is necessary in order to make the case that this construct is relevant to the present study. Self-esteem and self-concept are quite different constructs, and should be delineated accordingly.
- Table 1 could be improved by aligning specific entrepreneurship constructs with constructs from the other fields included in the table, rather than listing constructs in each column with no alignment across rows—and the Assessment in Education construct does not have any constructs listed, only references. A similar structure would benefit the manuscript itself, i.e., clarifying which constructs and outcomes in entrepreneurship are informed by psychological constructs, rather than listing psychological constructs and then discussing their relevance to entrepreneurship.
- The authors describe their work as a metaanalysis, but no metaanalysis results are presented. Please clarify.
- Figure 1 does not seem specific to entrepreneurship. It contains constructs present in established theories of self-regulation, and does not specify how it is specific to entrepreneurship or entrepreneurship education.
- Many of the conclusions do not align with evidence presented in the manuscript. For example, the sentence that starts with “Entrepreneurial individuals, having entrepreneurial competencies…” on Lines 508-512 does not seem based in any of the literature reviewed in the manuscript. Does any of the reviewed literature contain empirical evidence about the characteristics of entrepreneurial and non-entrepreneurial individuals? Most of the articles referenced appear to be about entrepreneurship education, not the quality of entrepreneurship itself.
Author Response
The authors make an important point regarding the importance of theoretical rationale for entrepreneurial competencies in addition to empirical evidence. Self-esteem and self-efficacy may be relevant contributors to such a rationale. Unfortunately, neither the arguments made nor literature reviewed are sufficient to constitute a theoretical rationale as intended. Overall, the authors have established a laudable goal, but they have not accomplished that goal in the present version of the manuscript. Specific suggestions can be found below.
Dear reviewer.
Thank you for your constructive feedback to develop our paper. Please, see our comments below which show how we have revised our paper.
- Numerous spelling errors make aspects of the manuscript difficult to understand, particularly towards the beginning (e.g., crusial, emphising).
Thank you. We have proofread the revised version of the paper.
2. The comparison of different self-efficacy theories would be clearer if it were formatted more like the summary that appears after each theory is described, rather than as a list of concepts for each theory separately.
Thank you for taking this challenging issue into account. During our writing process, we have also considered this. Finally, we came to this solution that we present in the paper, since another problem is that the concepts are interrelated, and at some point, we should integrate them. These different openings for examining these concepts would lead to the literature review becoming too broad for a single paper. However, we hope that this paper would be an opening for a more in-depth and detailed study of the theme in the future.
3. Some ideas are oversimplified or possibly misunderstood, e.g., Kolb wrote about “experiential” learning, not “experimental learning” (Table 1). Similarly, phrases like “Unlike in the psychology and medicine research, self-efficacy and self-esteem have arguably been the mostly researched constructs of people’s self-evaluations that are closely related also to their mental health” (Lines 363-365) are unclear.
Yes, we fully agree. We have corrected also the spelling errors. Furthermore, we have revised the unclear phrases throughout the paper.
4. Self-esteem is a controversial topic in psychology, because many studies have found that it is not linked to outcomes one might expect based on common understandings of the term. A more nuanced view of self-esteem is necessary in order to make the case that this construct is relevant to the present study. Self-esteem and self-concept are quite different constructs, and should be delineated accordingly.
Thank you. We have opened the concept of self-esteem in more detail (Chapter 2.2.). Furthermore, we have briefly discussed the differences between self-concept and self-esteem. However, it is difficult to extend their clarification since our main focus of this paper to discuss self-esteem and self-efficacy and their connections to each other.
5. Table 1 could be improved by aligning specific entrepreneurship constructs with constructs from the other fields included in the table, rather than listing constructs in each column with no alignment across rows—and the Assessment in Education construct does not have any constructs listed, only references. A similar structure would benefit the manuscript itself, i.e., clarifying which constructs and outcomes in entrepreneurship are informed by psychological constructs, rather than listing psychological constructs and then discussing their relevance to entrepreneurship.
Thank you for this excellent idea. Please, see our revised Table 1.
6. The authors describe their work as a metaanalysis, but no metaanalysis results are presented. Please clarify.
This a relevant notification. We have deleted this misleading concept from our study.
Figure 1 does not seem specific to entrepreneurship. It contains constructs present in established theories of self-regulation, and does not specify how it is specific to entrepreneurship or entrepreneurship education.
In our paper, we highlight how entrepreneurship education is about much more than just developing entrepreneurship. Entrepreneurship education is about entrepreneurial activity, which also involves self-regulation. The concept of entrepreneurship education can be often confusing - however, we use it in this context because the use of the term is justified in many studies and it has been extensively researched.
7. Many of the conclusions do not align with evidence presented in the manuscript. For example, the sentence that starts with “Entrepreneurial individuals, having entrepreneurial competencies…” on Lines 508-512 does not seem based in any of the literature reviewed in the manuscript. Does any of the reviewed literature contain empirical evidence about the characteristics of entrepreneurial and non-entrepreneurial individuals?
We have sharpen the use of literature references and the use of concepts. It should be noted, however, that we have discussed our results more broadly in the discussion and conclusion parts of the paper in which we both refer to the literature sources, and on the other hand, we also bring our own ideas related to the key points.
8. Most of the articles referenced appear to be about entrepreneurship education, not the quality of entrepreneurship itself.
We stress in our paper how to develop entrepreneurial competencies by entrepreneurship education. Therefore, our aim is not to consider the quality of entrepreneurship itself.
Reviewer 3 Report
A good abstract properly introduces the topic. Good theoretical section related to literature review. The impression is that Author(s) is well grounded in the subject of entrepreneurship/education. The Author(s) identifies the research gap with great skill.
I am not sure if it is a good idea to label section 2.2.3 as Summary. it is in the middle of the text and confuses the reader a bit. More clarity could be achieved by renaming section 2.2.3.
In my opinion, the last part “Conclusions” should be expanded. There is quite a big lack of harmony between the extended conceptual part and this conclusions. It seems that the conclusions should be supplemented with more discussion on the possibility of using the results of the theoretical research presented in this paper. Without an expanded conclusions, the article may not be adequately finished, especially since the entire text is the theoretical essay.
Author Response
A good abstract properly introduces the topic. Good theoretical section related to literature review. The impression is that Author(s) is well grounded in the subject of entrepreneurship/education. The Author(s) identifies the research gap with great skill.
Thank you.
I am not sure if it is a good idea to label section 2.2.3 as Summary. it is in the middle of the text and confuses the reader a bit. More clarity could be achieved by renaming section 2.2.3.
Thank you for the suggestion. We have revised this section.
In my opinion, the last part “Conclusions” should be expanded. There is quite a big lack of harmony between the extended conceptual part and this conclusions. It seems that the conclusions should be supplemented with more discussion on the possibility of using the results of the theoretical research presented in this paper. Without an expanded conclusions, the article may not be adequately finished, especially since the entire text is the theoretical essay.
We have expanded the conclusion from these points of views.
Thank you!
Round 2
Reviewer 2 Report
I appreciate the authors' attempt to revise the manuscript in accordance with reviewer feedback. However, I have two major remaining concerns that I am not sure the authors can address:
- The contribution of the construct of entrepreneurship education, beyond existing constructs in psychology such as self-efficacy and self-regulation, is unclear. In other words, what is entrepreneurship education, if it is not the training of entrepreneurs, other than the combination of several well-established psychological/educational constructs?
- The nuances of self-esteem vs. self-efficacy vs. self-regulation, and how they fit within entrepreneurship education, are still not well-described. There are remaining issues with the description of self-esteem, and the comparison of self-esteem to self-concept is not consistent with literature I am familiar with (e.g., "a realistic concept of self-image does not differ much from good self-esteem"). Seminal works in self-esteem the relationship between self-esteem and self-concept (e.g., Baumeister's work, Trautwein et al., 2006) should be reviewed to thoroughly explain these relationships.
Author Response
I appreciate the authors' attempt to revise the manuscript in accordance with reviewer feedback. However, I have two major remaining concerns that I am not sure the authors can address:
- The contribution of the construct of entrepreneurship education, beyond existing constructs in psychology such as self-efficacy and self-regulation, is unclear. In other words, what is entrepreneurship education, if it is not the training of entrepreneurs, other than the combination of several well-established psychological/educational constructs?
Thank you for challenging us with this issue. We have clarified and extended our definitions (p. 3, para. 6; p. 4, para 1; p. 9, para. 4)
- The nuances of self-esteem vs. self-efficacy vs. self-regulation, and how they fit within entrepreneurship education, are still not well-described. There are remaining issues with the description of self-esteem, and the comparison of self-esteem to self-concept is not consistent with literature I am familiar with (e.g., "a realistic concept of self-image does not differ much from good self-esteem"). Seminal works in self-esteem the relationship between self-esteem and self-concept (e.g., Baumeister's work, Trautwein et al., 2006) should be reviewed to thoroughly explain these relationships.
Thank you for the ideas. We have revised the literature review after exploring the suggested research in the field (p.5, para. 3,4; p. 6, para. 2,3,6)
We highly appreciate the referee´s efforts to improve our paper. Thank you again.
Round 3
Reviewer 2 Report
.